# Effect of Selected Stilbenoids on Human Fecal Microbiota

**DOI:** 10.3390/molecules24040744

**Published:** 2019-02-19

**Authors:** Jose D. Jaimes, Veronika Jarosova, Ondrej Vesely, Chahrazed Mekadim, Jakub Mrazek, Petr Marsik, Jiri Killer, Karel Smejkal, Pavel Kloucek, Jaroslav Havlik

**Affiliations:** 1Department of Food Quality and Safety, Czech University of Life Sciences Prague, Kamycka 129, 16500 Prague 6-Suchdol, Czech Republic; jose.d.jaimes@gmail.com (J.D.J.); jarosovaverca@gmail.com (V.J.); czeveselyo@gmail.com (O.V.); marsik@af.czu.cz (P.M.); kloucek@af.czu.cz (P.K.); 2Department of Microbiology, Nutrition and Dietetics, Czech University of Life Sciences Prague, Kamycka 129, 16500 Prague 6-Suchdol, Czech Republic; Chahrazedbiotek@gmail.com; 3Institute of Animal Physiology and Genetics, CAS, v.v.i., Videnska 1083, 14220 Prague, Czech Republic; kubino77@gmail.com (J.M.); killer.jiri@seznam.cz (J.K.); 4Department of Natural Drugs, Faculty of Pharmacy, University of Veterinary and Pharmaceutical Sciences Brno, Palackeho 1946/1, 61242 Brno, Czech Republic; karel.mejkal@post.cz

**Keywords:** phenolics, polyphenols, stilbenoids, human gut microbiota, 16S rRNA gene sequencing, batatasin III, oxyresveratrol, piceatannol, pinostilbene, resveratrol, thunalbene, fermentation, human colon model, *Lachnospiraceae*, Firmicutes, Bacteroidetes, *Clostridium*, *Faecalibacterium prausnitzii*

## Abstract

Dietary phenolics or polyphenols are mostly metabolized by the human gut microbiota. These metabolites appear to confer the beneficial health effects attributed to phenolics. Microbial composition affects the type of metabolites produced. Reciprocally, phenolics modulate microbial composition. Understanding this relationship could be used to positively impact health by phenolic supplementation and thus create favorable colonic conditions. This study explored the effect of six stilbenoids (batatasin III, oxyresveratrol, piceatannol, pinostilbene, resveratrol, thunalbene) on the gut microbiota composition. Stilbenoids were anaerobically fermented with fecal bacteria from four donors, samples were collected at 0 and 24 h, and effects on the microbiota were assessed by 16S rRNA gene sequencing. Statistical tests identified affected microbes at three taxonomic levels. Observed microbial composition modulation by stilbenoids included a decrease in the Firmicutes to Bacteroidetes ratio, a decrease in the relative abundance of strains from the genus *Clostridium*, and effects on the family *Lachnospiraceae*. A frequently observed effect was a further decrease of the relative abundance when compared to the control. An opposite effect to the control was observed for *Faecalibacterium prausnitzii*, whose relative abundance increased. Observed effects were more frequently attributed to resveratrol and piceatannol, followed by thunalbene and batatasin III.

## 1. Introduction

Stilbenoids are a subclass of plant-derived phenolic compounds often consumed in the diet as components from red grapes, peanuts, certain berries, and many others. Their average dietary intake is 1 g/day [1,2,3]. The most well studied stilbenoid is resveratrol, which came into the spotlight with the so-called French paradox, where it was attributed in reducing coronary heart disease mortality among the sample population despite the strong presence of risk factors [4,5]. Further studies have attributed many other potential health benefits to resveratrol, as well as to various other phenolics, such as potent antioxidant activity, cardio-protection, neuroprotection, anti-inflammatory effects, cancer prevention, and others [4].

In plants, phenolics are usually conjugated to sugars, organic acids, and macromolecules (e.g., dietary fiber and proteins) and most of them are not properly released and absorbed in the small intestine, reaching the colon for further microbial fermentation; at colonic level, they are fermented by the resident gut microbiota (GM) [6]. It is the resulting metabolites that are attributed the health benefits as bioactive compounds. Evidence shows that 90–95% of ingested dietary phenolics, usually in their glycosylated form, are not absorbed in the upper part of the digestive tract. Most of them reach the colon, where the GM metabolize them into lower molecular weight-phenolic compounds, such as phenolic acids, that can be more easily absorbed by intestinal epithelial cells and enter the liver for further biotransformation or systemic circulation [7,8,9,10,11,12,13]. These microbial bio-transformations are grouped into three major catabolic processes: hydrolysis (O-deglycosylations and ester hydrolysis), cleavage (C-ring cleavage; delactonization; demethylation), and reductions (dehydroxylation and double bond reduction) [14]. Reciprocal to these bio-transformations by the GM, phenolics appear to modulate the GM composition by favoring/disfavoring certain microbial strains, thus establishing a two-way relationship between the GM and phenolics [6,15,16,17,18]. The undigested phenolics, along with diet-independent substrates like endogenous host secretions, are the main substrates of gut bacterial metabolism, and may affect the GM in a similar manner as prebiotics, shape microbial composition by antimicrobial action, and/or influence bacterial adhesion [2,19,20,21,22,23,24]. For example, chlorogenic acid, resveratrol, catechin, and certain quercetin derivatives have exhibited prebiotic-like effects by increasing the proportional representation of *Bifidobacterium* strains [2,7,25,26,27]. Antimicrobial action has been shown by inoculation with resveratrol and certain ellagitannins by inhibiting the growth of several *Clostridia* species [2,12,17,28]. Bacterial adhesion effects by procyanidin and chlorogenic acid have been noticed through adhesion enhancement of certain *Lactobacillus* strains to intestinal epithelial cells [23,24].

To our knowledge, except for resveratrol and a few studies with piceatannol, both of which are well-recognized plant-derived phenolics, there is not much information regarding the effects of stilbenes on the GM. Other than a 2016 study on the effects of repeated stilbenoid administration on the GM, the rest have mostly focused on evaluating a single dose effect, mainly on culturable microbial strains. The findings from the former showed a strong change in the GM composition after application of resveratrol and viniferin, especially in the enrichment of the order Enterobacteriales, and a decrease of Bifidobacteriales [29]. Observations from the single dose studies showed changes in the GM composition; for example, increases for species *Akkermansia muciniphila* and *Faecalibacterium prausnitzii* by resveratrol, and in the genus *Lactobacillus* by piceatannol [2,7,8,27,30,31,32,33,34].

The objective of this study was to assess the effect of six stilbenoid phenolics (batatasin III (Bat), oxyresveratrol (Oxy), *trans*-resveratrol (Res), piceatannol (Pic), pinostilbene (Pino), and thunalbene (Thu); the corresponding chemical structures are given in Figure 1) on the GM at dietary relevant concentrations. Using an in vitro fecal fermentation (FFM) system, these stilbenoids were fermented with human fecal bacteria from four donors. Effects on the GM composition were based on 16S rRNA gene sequencing results.

## 2. Results and Discussion

### 2.1. Firmicutes to Bacteroidetes (F/B) Ratio

The most abundant phyla in human gut microbiota are Firmicutes and Bacteroidetes, which often account for more than 90% of the total gut microbiota [35]. However, that was not the case in this study. Firmicutes were the most abundant, followed by Actinobacteria, with Bacteroidetes coming in at either fourth or fifth place depending on the donor. One possibility may be that one of the kits used during processing may have been more sensitive to phyla other than Bacteroidetes, or perhaps these bacteria progress to a higher relative abundance during *in vitro* cultivation compared to what would normally be found in stool alone. Nevertheless, the ratio of these two phyla can still be evaluated.

An increased F/B ratio in both human and mouse gut microbiota has consistently been associated with higher obesity and disease occurrence [36,37]. Resveratrol has been previously shown to decrease this ratio [2,27,38], and our findings support this. Similarly, the other tested stilbenoids also decreased the F/B ratio as can be seen in Figure 2. Res, Bat, and Thu reached lower ratios (61 ± 23, 49 ± 22, 96 ± 53 respectively) than the control at 24 h (121 ± 73). Interestingly, Pino showed an increase (227 ± 127), while Pic stayed approximately equal (131 ± 98) to the control at 24 h. The response is a result of a decrease in the relative abundance of Firmicutes and an increase of Bacteroidetes, which is consistent with findings from other studies [2,7,27]. For Firmicutes, after treatment with all tested stilbenoids, the relative abundance decrease (−2.9% ± 0.03%) was lower than the control at 24 h (−4.6% ± 0.03%), with the least decrease observed under Oxy and Pino (−1.5% ± 0.03% and −0.7% ± 0.02%, respectively). For Bacteroidetes, after treatment with all tested stilbenoids except for Pino (51.0.2% ± 0.00%), the growth in relative abundance (Bat 278.0% ± 0.02%; Oxy 198.1% ± 0.00%; Pic 86.0% ± 0.05%; Res 195.6% ± 0.04%; Thu 300.3% ± 0.01%) was greater than that of the control at 24 h (68.0% ± 0.04%). 

### 2.2. Most and Least Abundant Species

A total of 230 bacterial species entities were detected in the tested fecal samples. This number includes unidentified species that could only be categorized as part of a higher taxonomic level. For example, an unidentified species, from an unidentified genus, that belongs to the *Clostridiaceae* family. The lowest detected relative abundance was 0.00047% for an unidentified species of the *Christensenella* genus.

The five species with the highest relative abundance per each of the tested samples were identified. These accounted for 53% to 66% of the total relative abundance and, in total, comprised 11 distinct species (Table 1). Therefore, there appears to be certain consistency, and not much variability, among the most abundant taxa.

Focusing on the inverse, in the five species with the least relative abundance, there is less consistency and greater variability since it comprised 27 distinct species (Appendix A, Table A1). It’s important not to ignore the least abundant species since their low abundance may not necessarily correlate with the importance of their function. As stated in Cueva et al., the microorganisms present in smaller quantities, but developing specific functions, could be the key to understanding the individual response to consumption of bioactive compounds (i.e., phenolics). Some metabolic functions seem to be achieved by a wide variety of species, while other functions are only done by a specific few [15]. For example, *Ruminococcus bromii*, identified within the 27 species, has been noted to be a butyrate (a short-chain fatty acid) producer, which is a function that appears to be found in fewer species than those for acetate [39].

### 2.3. Changes in Relative Abundance (Phylum, Family, Species)

Both parametric and non-parametric statistical tests were used to identify taxa of interest at the phylum, family, and species level based on two comparisons. The statistical tests were used as a tool to identify potential significantly affected taxa, and should not be interpreted as a portrayal of definite statistical significance (for those with p values in the range) due to the small sample size (four donors). The identified taxa reported p < 0.075 for at least one *p* value (paired sample t-test and/or Wilcoxon signed-rank test) for both comparisons 1 and 2. Comparison 1 used as a baseline the relative abundance of the 24 h control, and compared this value to each of the six stilbenoid fermentations. Comparison 2 used as a baseline the magnitude of change (growth or decline) in relative abundance between Control 0 h and Control 24 h, and compared this value to the magnitude of change between Control 0 h and each of the six stilbenoid fermentations. 

Figure 3 displays these identified taxa in the form of a phylogenetic tree sorted by phylogenetic distance. The corresponding *p* values are listed in Appendix A, Table A2, and the corresponding relative abundance box plots are shown in Figure 4. Each comparison (1&2) is shown separately in Appendix A, Table A3 and Table A4, and list additional taxa. Clustered bar graphs of bacterial composition at the phylum and family levels can be seen in Appendix A, Figure A1 and Figure A2. Table 2 displays how our study compares to findings and observations from other studies regarding the effect of the selected stilbenoids on a specific taxon. 

#### 2.3.1. Decrease in Relative Abundance

A decrease in relative abundance was observed for several taxa under some of the tested stilbenoids. The most frequently observed response was a further decrease of the relative abundance of a specific taxon as compared to the 24 h control by either Res, Pic or Thu. For example, for *Clostridium sp.* there was a decrease of −54.2% ± 28.8% for Ctrl24, while the decrease caused by Pic and Thu were of a greater magnitude, −62.9% ± 28.0% (t(3) = 3.960, p = 0.029) and −79.3% ± 22.6% (t(3) = 3.901, p = 0.030), respectively. Similar responses were observed, albeit at different magnitudes, for family *Lachnospiraceae,* and species *Coprococcus sp.*, *Collinsella aerofaciens,* and *Lachnospiraceae Gen. sp*. At the genus level, *Clostridium* decreased under all tested stilbenoids in our study. Previous findings, as listed in Table 2, observed that several species from the genus *Clostridium*, which includes both commensal and deleterious species, had been shown to decrease with resveratrol [2,12].

A second observed response was a decrease in relative abundance while the 24 h control increased. This was observed by three species, *Ruminococcus sp.* (−3.2% ± 69.1%, t(2) = 4.448, p = 0.047 under Bat; −7.0% ± 69.4%, t(3) = 8.253,p = 0.004 under Pic; −41.1% ± 50.9%, t(3) = 1.953, p = 0.146 under Thu)*, Ruminococcus sp.* (−3.3% ± 12.7%, t(3) = 3.947, p = 0.029 under Res), and *Coriobacteriaceae Gen. sp.* (−0.9% ± 94.2%, t(2) = 6.272, p = 0.024 under Oxy; −3.7% ± 90.6%, t(3) = 3.261,p = 0.047 under Pic; −39.2% ± 10.0%, t(3) = 1.726, p = 0.183 under Thu), while they increased in the 24 h control (27.8% ± 80.6%, 32.2% ± 68.5%, 15.5% ± 20.8%, respectively). Regarding *Ruminococcus,* this may not be a favorable response according to recent research that points to a high proportion of long-chain dietary fibers degraders, butyrate producing bacteria such as *Ruminococcus, Eubacterium, and Bifidobacterium* as being part of healthy gut microbiota [11,40,41,42]. The *Ruminococcus* genus has previously been identified as one of the three taxa, besides *Bacteroides* and *Prevotella*, that define the enterotype concept, which could help in explaining variability in responders/non-responders in intervention studies [43]. In regards to *Coriobacteriaceae*, it has been noted that many species that metabolize phenolics belong to this family, however, its potential health implications are still poorly understood [6]. Nevertheless, one important aspect of this family is that all identified *S*-equol-producing bacteria, except for the genus *Lactococcus,* belong to it [44,45].

A third observed response was a decrease in relative abundance while the 24 h control also decreased, but with a larger magnitude. This was observed for *Blautia obeum*, which was recently reclassified, its former name being *Ruminococcus obeum* [46]. *Blautia* has been considered one of the major representatives of the Firmicutes phylum due to its relatively high abundance [15]. This species experienced a decrease in relative abundance by thunalbene (−5.6% ± 32.1%, (t(3) = 3.763, p = 0.033), but at a lower magnitude than the control at 24 h (−29.8% ± 35.6%). A decrease of *Blautia,* at the genus level, was also reported in a study conducted on mice fed a phenolic-enriched tomato diet, as well as in a study of human fecal fermentation study after consumption of phenolics from tart cherries [26,31]. These findings, along with our study, suggest that certain phenolics may cause a decrease in this genus, but at a lesser magnitude than without it. This taxon also appears to be a butyrate-producing microbe whose reduction has been correlated with decreased production of butyrate [47]. 

Eight of the identified taxa belonged to the family *Lachnospiraceae*. There was no consistent response from the tested stilbenoids within this family however, the most frequent response was a decrease in relative abundance. This decrease was also observed in a study where rats were supplemented with the stilbenoid pterostilbene in their diet. In that study, *Lachnospiraceae* was significantly reduced in each tested group when compared to baseline levels [33].

#### 2.3.2. Increase in Relative Abundance

An increase in relative abundance with no change in the 24 h control was observed for *Faecalibacterium prausnitzii* under Res (36.6% ± 88.0%, t(3) = −2.806, p = 0.068 under Res), 24 h control (−0.5% ± 62.5%). This species has been previously identified as a butyrate producing bacterium and is regarded as being beneficial. Butyrate production appears to be key in maintaining the colonic epithelium by inducing proliferation of healthy colonocytes. Fiber-poor diets, such as the one our donors were subject to prior to sample donation, have been associated with low butyrate production. One study showed a strong positive correlation between the proportion of *F. prausnitzii* and that of butyrate in individuals on a normal diet, and the reduction in *F. prausnitzii* on switching to a fiber-free or fiber-supplemented diet correlated with the reduction in fecal butyrate [47,48]. The gut epithelium is the main body site for butyrate sequestration, and low butyrate production has been connected to inflammatory diseases such as ulcerative colitis [39,49]. Unlike acetate producing bacteria, which are widely distributed, there appear to be fewer butyrate producing bacteria such as *S. prausnitzii, E. rectale, E. hallii,* and *R. bromii* [38]. It was observed to increase in plant-based, fiber-rich, diets, thus, stilbenoids being phytochemicals, were expected to increase their abundance. Our findings support this with resveratrol.

An increase in relative abundance with a decrease in the 24 h control was observed for *Ruminococcus gnavus* under Thu (8.2% ± 40.6%, t(3) = −2.244, p = 0.111 under Thu), 24 h control (−12.9% ± 30.7%). The observed p value, along with the box plot in Figure 4, show that *R. gnavus’* increase was not as pronounced as that of *F. prausnitzii*. Both of these taxa tend to be quite reduced in inflammatory bowel diseases such as Crohn’s disease [50,51].

Although it was detected in only one of our donors, *Akkermansia muciniphila* was observed to be enhanced by resveratrol. This species has been previously observed to be enhanced by pterostilbene, which has shown to exhibit similar cellular effects to resveratrol. One of these is that both phenolics have been hypothesized to mimic caloric restriction effects at the molecular level, thus modifying the gut microbiota, especially enhancing *A. muciniphila* [33]. 

These findings emphasize the importance of trying to get to the lowest possible taxonomic level to better characterize the gut microbiota. As can be seen from our study, species within the same family level are not all uniform in their responses. Higher taxonomic levels are quite useful, and can make experiments and data processing much more manageable; however, care must be taken in generalizing for every member of a taxon.

Whether the microbiota response is a decrease or an increase in relative abundance, effects are more frequently attributed to resveratrol and piceatannol, followed by thunalbene and batatasin III. This difference may be related to their chemical moieties. All stilbenoids share a basic C6-C2-C6 structure, differing only in the presence or absence of a C-C double bond on -C2-, and on the type and position of functional groups, mainly hydroxyl (-OH) and o-methoxyl (-OCH_3_) groups on the aromatic rings. In phenolics, -OH groups play an important role on their bioactivity, and their substitution by -OCH_3_ groups has been shown to reduce their bioactivity [52,53,54]. -OH groups are good hydrogen donors, are considered very reactive and potent radical scavengers, are key in the general antioxidant mechanism of resveratrol, and it has been shown that phenolics with more -OH groups exhibit higher capacity for enzyme inhibition than those with -OCH_3_ groups [53,54,55,56,57]. Enzyme inhibition capacity has also been shown to be affected by hydrogenation of the C-C double bond on -C2-, which decreased enzyme inhibition [54,58,59,60]. This suggests that phenolics with -OH moieties and C-C double bond on -C2- may be more bioactive than those with -OCH_3_ moieties and lacking a C-C double bond on -C2-. Resveratrol and Piceatannol have three and four -OH groups respectively, as well as a C-C double-bond on -C2-. They were the two stilbenoids that were most frequently attributed effects on the GM in this study. These were followed by thunalbene, which is O-methylated and has a C-C double bond on -C2-, and by batatasin III, which is O-methylated and lacks a C-C double bond on -C2-. Regarding demethylation, a recent study reported a demethylated colonic metabolite of the phenolic curcumin by *Blautia sp.* MRG-PMF1 [61]. Thunalbene is O-methylated and, as reported earlier, *Blautia sp.* experienced a decrease in relative abundance under thunalbene, but at a lower magnitude than that of the control. Regarding C-C double bond reduction, Bode et al. showed that *Slackia equalifaciens* and *Adlercreutzia equolifaciens* were able to metabolize resveratrol to dihydroresveratrol by reduction of the C-C double bond, but could not identify any bacteria for the -OH cleavage that produced two other metabolites [8]. Reduction of the C-C double bond by GM has also been shown for other phenolics such as isoflavones and hydroxycinnamates, while -OH cleavage for lignans and phenolic acids [19,62,63,64]. How chemical moieties affect metabolite production by microbial strains and bioactivities such as antioxidant activity, enzyme inhibition, quorum sensing, and others is outside the scope of our study; nevertheless, it’s an important avenue for ongoing and future research. 

The interpretation of the results from GM studies such as this one should take into consideration the concept of inter-individual variability. This concept is well known in the literature, the most well-known example being the difference between individuals whose microbiota are either producers or non-producers of the *S*-equol phytoestrogen. Oral administration of *S*-equol results in improvement of certain cardiovascular disease biomarkers, but only on those who are producers [20,47]. Although our sample size is small, differences among donor GM composition can be visualized in Figure A1 and Figure A2. Donor D2, for example, appears to have a very atypical microbial composition when compared to the other three donors. 

## 3. Materials and Methods 

### 3.1. Study Design

Using an in vitro fecal fermentation (FFM) system, a set of six stilbenoid phenolics were fermented in vials via inoculation with human fecal bacteria obtained from four donors. The vials were sampled at 0 hour (0 h) and 24 h (24 h) time points, and the effect of the stilbenoids on human GM was assessed by 16S rRNA gene sequencing. Both parametric and non-parametric statistical tests were used to identify potentially affected strains at the phylum, family, and species taxonomic levels.

### 3.2. Donors and Ethics Statement

The fecal samples originated from four volunteer donors, all of whom consented for their samples to be used for research purposes by signing a consent form. The ethical agreement for stool collection was obtained by the ethical committee (ZEK/22/09/2017) of the Czech University of Life Sciences in Prague. The donors were two males and two females ages 23, 28 (Donors 1 and 3) and 26, 29 (Donors 2 and 4) respectively. Their respective body mass index (BMI) were 23.0, 24.7, 26.0, and 26.5. To reduce potential interference from other dietary phenolics, all donors followed a low-phenolic diet for at least 48 hours prior to providing the fecal sample. Also, none had taken any antibiotics for at least 6 months prior to sampling. They described themselves as being in good health, and none reported any chronic conditions or diseases. They followed an omnivorous diet in their daily life. Females were neither pregnant nor lactating. The samples were collected in October and November 2016, at the Czech University of Life Sciences in Prague, Czech Republic.

### 3.3. In vitro Fecal Fermentation (FFM) System

#### 3.3.1. Standard Compounds and Chemicals

The chemicals used for preparation of the fermentation medium were obtained from Merck (Darmstadt, Germany). The stilbenoids batatasin III, piceatannol, thunalbene, and pinostilbene were purchased from ChemFaces (Wuhan, China) in 98% purity; *trans*-resveratrol, oxyresveratrol were obtained from Merck in 98% purity. Standards were prepared as 1% methanol/formic acid. Methanol and ethyl acetate were of analytical grade and purchased from VWR Chemicals (Stribrna Skalice, Czech Republic). Dimethyl sulfoxide (DMSO) was obtained from VWR Chemicals. Formic acid was obtained from Fisher Scientific (Merelbeke, Belgium) in >98% purity. Ultra-pure water (MilliQ) was obtained from a Millipore system (Bedford, MA, USA).

#### 3.3.2. Fermentation Medium

Fermentation medium was prepared from the following solutions based on previous fecal fermentation studies [66,67,68]. Micromineral solution was prepared from 2.64 g CaCl_2_, 2 g MnCl_2_, 0.2 g CoCl_2_, 1.6 g FeCl_3_, and up to 20 mL distilled water. Macromineral solution was prepared from 7.14 g of Na_2_HPO_4_, 6.2 g KH_2_PO_4_, 0.6 g MgSO_4_, and up to 1 L distilled water. Carbonate buffer was made of 1 g NH_4_HCO_3_, 8.75 g NaHCO_3_, and distilled water up to 250 mL (stored max. 1 month). The fermentation medium was prepared from 225 mL distilled water and 1.125 g of tryptone, 56.25 μL of micromineral solution, 112.5 mL of CO_3_ buffer, 112.5 mL of macromineral solution, and 562.5 μL of 0.1% resazurin solution.

#### 3.3.3. Phosphate Buffer, Reducing Solution

Sodium phosphate buffer for the preparation of fecal slurries was made of 1.7702 g KH_2_PO_4_ in distilled water (195 mL), and 3.6222 g Na_2_HPO_4_ in 305 mL distilled water (both 1/15 M). Afterwards, the buffer’s pH was modified to 7.0 by hydrochloric acid. Reducing solution was prepared from 125 mg cysteine hydrochloride, 0.8 mL 1 M NaOH, 125 mg Na_2_S and distilled water up to 20 mL.

#### 3.3.4. Fermentations Using Human Fecal Microbiota

Each tested stilbenoid was dissolved in DMSO to reach a concentration of 10 mg/mL. The fermentation medium and sodium phosphate buffer were boiled and cooled to approximately 37 °C while they were purged with oxygen free nitrogen gas (approx. 30 min). The medium’s pH was adjusted to pH 7.0 using HCl. For each vial, 16.8 mL of medium was transferred to the corresponding fermentation bottle and 0.8 mL of reducing solution was added. Per each donor, freshly obtained feces were homogenized in a stomacher bag with the sodium phosphate buffer to make a 32% fecal slurry. This slurry was then filtered through a mesh, from which 2 mL of the resulting filtrate was mixed with the fermentation medium in each of the fermentation bottles. 20 μL of tested compound solution (or DMSO alone for the controls) was also added. The bottles were incubated at 37 °C for 48 hours in a shaking bath at 100 strokes per minute. Four aliquots of fecal suspensions were prepared in 1.5 mL Eppendorf tubes by transferring from 20 mL glass bottles, collected at 0, 2, 4, 8, 24 and 48 h, and stored at −80 °C until further analysis. These timepoints were used for a related metabolomic study. For this particular study, only 0 and 24 timepoints were used.

### 3.4. Microbial Analysis

#### 3.4.1. DNA Extraction

Bacterial DNA was isolated from the fecal samples according to the manufacturer’s instructions using the Quick-DNA Fecal/Soil Microbe Miniprep Kit (Zymo Research, Irvine, CA, USA). The purified DNA was eluted in 100 µL of elution buffer and stored at −20 °C until further use.

#### 3.4.2. 16.S rDNA amplification: Nested PCR

During this nested PCR, two genes were amplified and targeted by two different pairs of primers in two successive reactions of PCR. The first PCR was done to amplify almost full length bacterial 16S rRNA gene fragments using the universal bacterial primers 616V (5′(5′ AGA GTT TGA TYM TGG CTC 3′) and 630R (5′ CAK AAA GGA GGT GAT CC 3′) [69]. The thermal cycling was carried out with an initial denaturation step of 94 °C for 5 min, followed by 32 cycles of denaturation at 94 °C for 45 s, annealing at 52 °C for 1 min, and elongation at 72 °C for 1 min and 30 s; cycling was completed by a final elongation step of 72 °C for 6 min. Using the purified PCR product from the first PCR, the second PCR was performed as described by Fliegerová et al. [70] to amplify the V4-V5 region of the 16S rRNA gene by the primer pair: BactBF (GGATTAGATACCCTGGTAGT) and BactBR (CACGACACGAGCTGACG). The used thermal cycling program was: initial denaturation for 5 min at 95 °C, followed by 35 cycles of 30 s at 95 °C, 30 s at 57 °C and 30 s at 72 °C, ending by final elongation for 5 min at 72 °C. The PCR amplicons (300 bp) were checked at 1.5% agarose electrophoresis (30 min at 100 V), purified by QIAquick PCR Purification Kit (Qiagen, Venlo, The Netherlands) according to the protocol and quantified by Nanodrop (Thermo Fisher, Waltham, MA, USA).

#### 3.4.3. Semi-conductor Based Next Generation Sequencing

Obtained PCR products were used to prepare libraries for diversity analyses by next generation sequencing (NGS) approach on Personal Genome Machine (Life Technologies, Carlsbad, CA, USA) according to Milani et al. [71]. 200 ng of DNA from each sample was used to prepare sequencing libraries by NEBNext® Fast DNA Library Prep Set kit (New England Biolabs, Ipswich, MA, USA) according to manufacturer’s protocol. The Ion Xpress Barcode adapters (Thermo Fisher Scientific, Waltham, MA, USA) were used to label each sample. The adaptor ligated libraries were purified and simultaneously size-selected using AMPure XP bead sizing (Beckman Coulter, Brea, CA, USA). The barcoded libraries were pooled in equimolar amount (about 26 pM). The pool of libraries was used to prepare sequencing template by emulsion PCR on Ion Sphere Particles (ISPs) using Ion PGMTM Hi-QTM View OT2 400 Kit (Thermo Fisher Scientific) in Ion OneTouchTM 2 instrument. The enrichment of the template positive ISPs were performed on Ion OneTouchTM ES instrument. The enriched template positive ISPs were then loaded in Ion 316TM Chip v2 BC (Thermo Fisher Scientific). The sequencing was then performed on an Ion Torrent PGM sequencer (Thermo Fisher Scientific, Waltham, MA, USA) using Ion PGMTM Hi-QTM View Sequencing solutions kit (Thermo Fisher Scientific).

#### 3.4.4. Data Analysis

The sequences obtained in FASTQ format were processed by QIIME analyses pipeline [72]. The chimeras were removed by USEARCH tool [73]. Remaining sequences were clustered and identified by performing open-reference OTU picking against the Greengene database [74]. Diversity index analysis and unweighted and weighted UniFrac distance metrics analyses were generated using QIIME and expressed by principle coordinate analysis (PCoA).

### 3.5. Statistical Analysis

Using SPSS version 25 (IBM Corp., Armonk, NY, USA), both parametric and non-parametric statistical tests were used to identify taxa of interest at the phylum, family, and species level by the following comparisons: (1) Using the relative abundance of the control fermentation with stool samples at 24 h with DMSO only as our baseline for comparison, we identified taxa from the fermentations with stilbenoids (each comparison done separately) that had *p* values <0.05 for the Paired sample t-test, and/or <0.075 for the Wilcoxon signed-rank test when compared to our baseline. (2) The magnitude of change (growth or decline) in relative abundance between the control fermentation with only stool sample at 0 h (Ctrl0 h) and our control fermentation with samples with DMSO only at 24 h was calculated, and this value became our baseline for comparison against the magnitude of change from 0 h to 24 h for the fermentations with stilbenoids. Selected taxa had *p* values <0.05 for the Paired sample t-test, and/or <0.075 for the Wilcoxon signed-rank test. Only 5 stilbenoids were tested. Pinostilbene was excluded since samples for it were only available for two out of the four donors. Similarly, any pair that had n ≤ 2 was excluded. Since the data was in percent, the magnitude of change was obtained by obtaining the percentage change of the given percentages. Values of 0% at 0 h were excluded, even if they were detectable at higher percentages. This was done due to the ambiguity of whether they were low values that were undetectable or whether they were simply not present.

## 4. Conclusions

From the surveyed literature, none of the tested stilbenoids, other than resveratrol and piceatannol, had been tested on their effect on the human GM. Our findings suggest that the tested stilbenoids, at physiological concentrations of 10 µg/mL, modulate the GM as observed in a fecal fermentation human colon model. Some of these effects are similar to other studies that have also assessed the effects of dietary phenolics on the GM. Some of our observed effects include a decrease in the Firmicutes to Bacteroidetes ratio, a consistent decrease in the relative abundance of strains from the genus *Clostridium*, and responses from several strains from the family *Lachnospiraceae*. A frequently observed effect on the identified taxa was a further decrease of the relative abundance when compared to the control. An opposite effect to the control was observed for *Faecalibacterium prausnitzii*, which, contrary to the control, increased in relative abundance. This strain has been previously considered beneficial for health. Looking at specific stilbenoids, observed responses were more frequently attributed to resveratrol and piceatannol, followed by thunalbene and batatasin III.

The use of 16S rRNA gene sequencing, in combination with a fecal fermentation human colon model, appears to be a very useful tool to characterize the human GM, especially to identify unculturable strains. It is important to note that studies such as this one are expected to increase in precision as the sensitivity of the detection technology, as well as the taxonomical reference databases, are refined and expanded. The tested stilbenoids appear to support the well-observed view of the potential positive impact of phenolics through the modulation of human GM, and thus further studies are recommended to characterize this microbial environment and its function more precisely. 

## Figures and Tables

**Figure 1 molecules-24-00744-f001:**
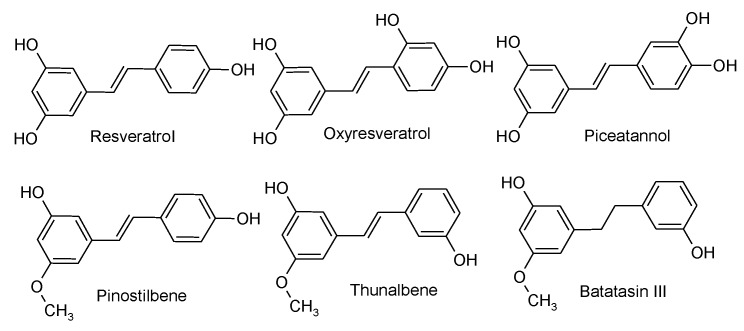
Molecular structures of stilbenoids studied. All stilbenoids have a C_6_-C_2_-C_6_ structure.

**Figure 2 molecules-24-00744-f002:**
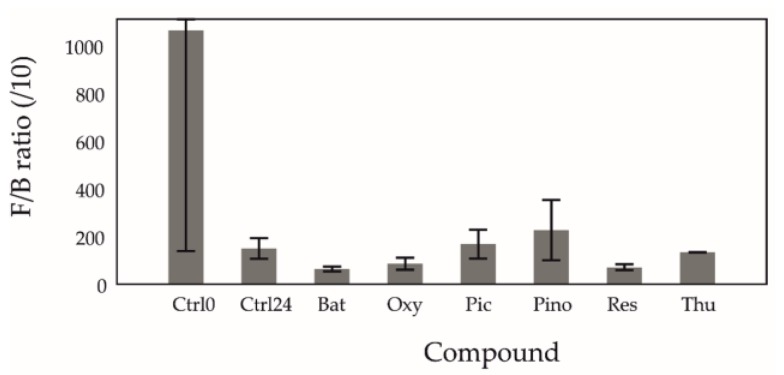
Mean Firmicutes/Bacteroidetes ratio (/10) in fermentations. Error bars represent the 95% CI. Ctrl0 = control at 0 h; Ctrl24 = control at 24 h; Bat = batatasin III; Oxy = oxyresveratrol; Pic = piceatannol; Pino = pinostilbene; Res = *trans*-resveratrol; Thu = thunalbene. All stilbenoids at 24 h.

**Figure 3 molecules-24-00744-f003:**
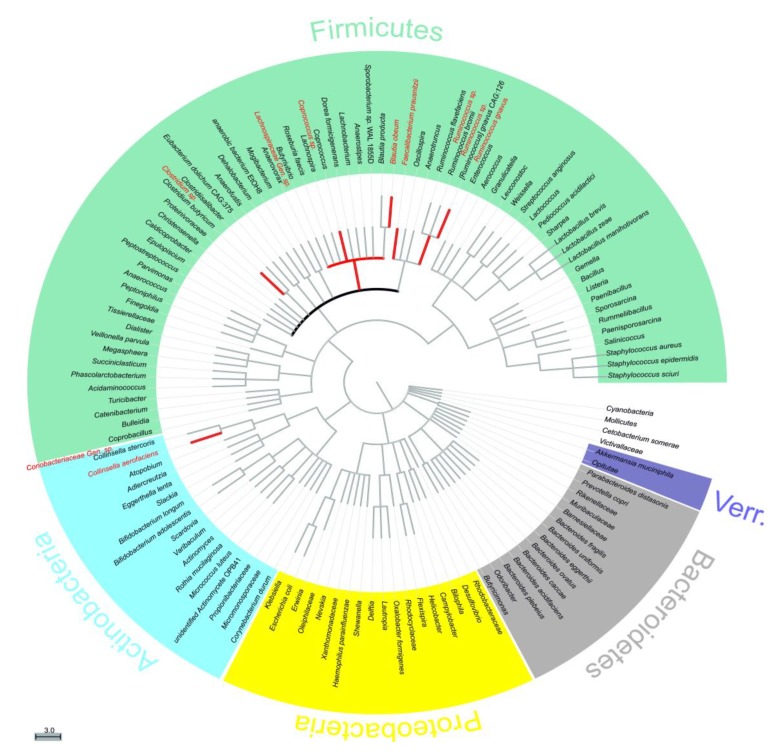
Phylogenetic tree of all identified bacterial entitites. The tree is sorted by phylogenetic distance, therefore the closer they are on the tree, the closer they are genetically [41,42]. Taxa shown in red displayed p < 0.075 for at least one p value (Paired sample t-test and/or Wilcoxon signed-rank test) for both comparisons 1 and 2. Bolded black line refers to family Lachnospiraceae. Verr. = Verrucomicrobia. Gen. = unnamed genus, sp. = unnamed species.

**Figure 4 molecules-24-00744-f004:**
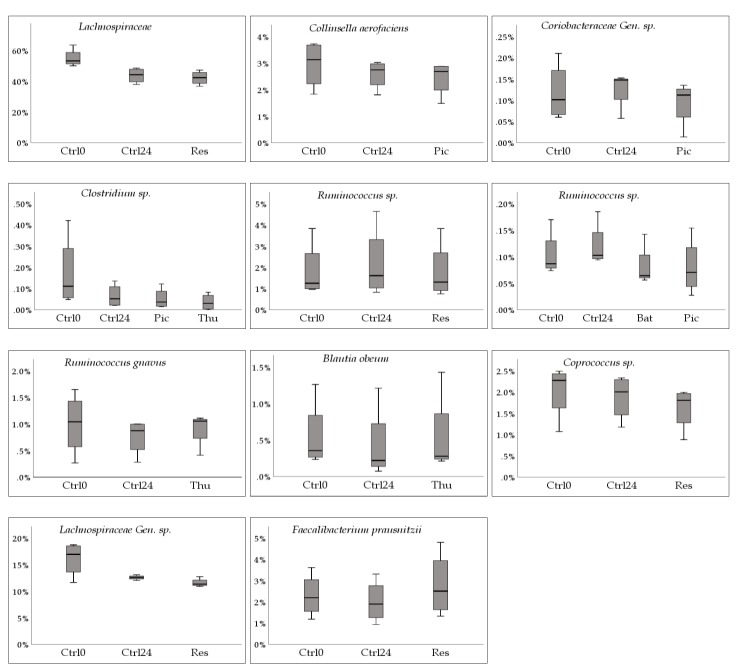
Box plots corresponding to the identified species and stilbenoids in Table A2. The y-axis displays relative abundance (%). The x-axis shows Ctrl0, control at 0 h; Ctrl24, control at 24 h; as well as the stilbenoid(s) corresponding to the observation (Bat, batatasin III; Oxy, oxyresveratrol; Pic, piceatannol; Pino, pinostilbene; Res, trans-resveratrol; Thu, thunalbene (all stilbenoids at 24 h). Gen. = unnamed genus, sp. = unnamed species.

**Table 1 molecules-24-00744-t001:** The most abundant species obtained by identifying the five species with the highest relative abundance for Control 0 h and 24 h, and per each of the six tested stilbenoid samples at 24 h. Gen. = unnamed genus, sp. = unnamed species.

Phylum	Class	Order	Family	Genus	Species
Actinobacteria	Actinobacteria	Bifidobacteriales	*Bifidobacteriaceae*	*Bifidobacterium*	*sp.*
Firmicutes	Bacilli	Lactobacillales	*Streptococcaceae*	*Streptococcus*	*sp.*
	Clostridia	Clostridiales	*Lachnospiraceae*	*Blautia*	*sp.*
				*Gen.*	*sp.*
				*Gen.*	*sp.*
			*Ruminococcaceae*	*Faecalibacterium*	*prausnitzii*
				*Ruminococcus*	*sp.*
				*Gen.*	*sp.*
				*Gen.*	*sp.*
			*Unnamed*	*Gen.*	*sp.*
Proteobacteria	Gammaproteobacteria	Enterobacteriales	*Enterobacteriaceae*	*Gen.*	*sp.*

**Table 2 molecules-24-00744-t002:** Observations from previous studies regarding the effect of select stilbenoids on specific taxa compared to observations in this study [2,7,8,27,30,31,32,33,34,65]. From the literature, ↑ or ↓ indicate a reported abundance increase or decrease of the strain. From this study, S, NS, Un, ND, signify, respectively, supported, not supported, undefined, not detected. Gen. = unnamed genus, sp. = unnamed species.

Stilbenoid	Effect	Phylum	Family	Genus	Species	Notes	
Resveratrol	↑	Actinobacteria	*Bifidobacteriaceae*	*Bifidobacterium*	*sp.*	NS	
		Firmicutes	*Clostridiaceae*	*Clostridium*	*XB90*	S	
				*Faecalibacterium*	*prausnitzii*	S	Won’t grow without acetate in pure culture.
			*Lactobacillaceae*	*Lactobacillus*	*sp.*	Un.	
	↓	Bacteroidetes	*Tannerellaceae*	*Parabacteroides*	*distansonis*	NS	Only detected in one donor.
		Firmicutes	*Clostridiaceae*	*Clostridium*	*aldenense*	S	Species not identified, responsesignificn at genus level.
					*C9*	S	
					*hathewayi*	S	
					*MLG661*	S	
			*Enterococcaceae*	*Enterococcus*	*faecalis*	ND	
			*Gracilibacteraceae*	*Gracilibacter*	*thermotolerans*	ND	
		Proteobacteria	*Enterobacteriaceae*	*Proteus*	*mirabilis*	ND	
		Firmicutes to Bacteroidetes (F/B) ratio				S	
	Other	Actinobacteria	*Coriobacteriaceae*	*Slackia*	*equolifaciens*	Other	Dihydroresveratrol producers. Identified at genus level only. Slackia’s abundance highest for Res, and not detectable at Ctrl0. Adlercreutzia’s abundance highest for Ctrl24, and lowest for Ctrl0.
				*Adlercreutzia*	*equolifaciens*	Other	
Phenolic mix, includes Resveratrol	↑	Verrucomicrobia	*Verrucomicrobiaceae*	*Akkermansia*	*muciniphila*	S	Mice study. Detected in one of our donors.
	↓	Firmicutes	*Lachnospiraceae*	*Blautia*	*sp.*	Un.	Mice study.
			*Ruminococcaceae*	*Oscillospira*	*sp.*	S	Mice study. Has never been cultured, but always detected.
Piceatannol	↑	Firmicutes	*Lactobacillaceae*	*Lactobacillus*	*sp.*	NS	Mice study.
			*Unnamed*	*Gen.*	*sp.*	NS	Mice study.
	↓	Bacteroidetes	*Unnamed*	*Gen.*	*sp.*	NS	Mice study. Decrease was observed, but at a lower magnitude than Ctrl24.
	Other		*Bacteroidaceae*	*Gen.*	*sp.*	S	Mice study. Abundance change.
Fiber	↑	Bacteroidetes	*Prevotellaceae*	*Prevotella*	*sp.*	S	Stilbenoids associated with fiber-containing food.
Plant-based diet		Firmicutes	*Clostridiaceae*	*Faecalibacterium*	*prausnitzii*	S	Saccharolytic microbes.
			*Lachnospiraceae*	*Roseburia*	*sp.*	NS	Saccharolytic microbes.
	↓	Proteobacteria	*Desulfovibrionaceae*	*Bilophila*	*sp.*	ND	Putrefactive microbes. Less abundance expected in a plant-based diet.
		Bacteroidetes	*Bacteroidaceae*	*Bacteroides*	*sp.*	NS	Putrefactive microbes. Less abundance expected in a plant-based diet.

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
