# Peer review of "Effect of Selected Stilbenoids on Human Fecal Microbiota"

_molecules, 2019, doi:10.3390/molecules24040744_

Round 1

Reviewer 1 Report

The referee would agree with publication of this manuscript after consideration described below.

lines 68-71: Among the six tested stilbenoids in the manuscript, it is well-recognized that  resveratrol and piceatannol, which are previously reported similar study,  are one of the naturally occuring polyphenols.

Why are you comparing this effects of the other minor stilbenoids with those of resveratrol and piceatannol? Please describe the background of the idea of theis research in more detail.

Do you have the evidence that resveratrol and piceatannol are metabolized to the other minor stilbenoids by this numan gut microbiota?

The chemical structure of six stilbenoids in Appendix 1 should be indicated in the main text.

Figure A2: Among the four donors, the desult of the bacterial composiotn of D2 showed a quite different with those of the others. Please describe the consideration on this result.

Author Response

Comment: lines 68-71: Among the six tested stilbenoids in the manuscript, it is well-recognized that  resveratrol and piceatannol, which are previously reported similar study,  are one of the naturally occurring polyphenols.

Response: We have reworded the sentence to express that resveratrol and piceatannol are well-recognized naturally occurring polyphenols. The new statement is: “To our knowledge, except for resveratrol and a few studies with piceatannol, both of which are well-recognized plant-derived phenolics, there is not much information regarding the effects of stilbenes on the GM.”

Comment: Why are you comparing this effects of the other minor stilbenoids with those of resveratrol and piceatannol? Please describe the background of the idea of theis research in more detail.

Response: We are not comparing the effect of minor stilbenoids to resveratrol and piceatannol, but we are attempting to increase the knowledge of the effects of this phenolic subclass (stilbenoids). Part of our aim was to explain other effects of potent inhibitors of cyclooxygenase (COX) enzymes previously found in another study https://www.sciencedirect.com/science/article/pii/S0308814619302031

We mentioned resveratrol and piceatannol only as a point of reference since not much is known about the other stilbenoids. In this study we are focusing on their effect on the human gut microbiota. Concurrently, using the same fecal samples, we are also conducting another study in regards to fecal bacteria derived metabolites from our tested stilbenoids.  

Comment: Do you have the evidence that resveratrol and piceatannol are metabolized to the other minor stilbenoids by this numan gut microbiota?

Response: A related metabolomic study (lines 368-369) from our lab team focuses more on identifying fecal bacteria derived metabolites from our tested stilbenoids, and we do see evidence of metabolites from resveratrol and piceatannol. However, since the results from the metabolomic study have not yet been published, and in order to keep this paper focused, we decided to direct this manuscript specifically on the effects of stilbenoids on the human gut microbiota. Nevertheless, we do cite in our manuscript (lines 291-293) a study by Bode et al. where three resveratrol metabolites were identified: dihydroresveratrol, 3,4’-dihydroxy-trans-stilbene and 3,4’-dihydroxybibenzyl (lunularin). Regarding piceatannol, we did not find any information regarding its metabolites, but our metabolomic study, which will refer to this paper, may shed more light on this.

Comment: The chemical structure of six stilbenoids in Appendix 1 should be indicated in the main text.

Response: The figure showing the chemical structures of the six stilbenoids was moved to the main text at the end of the introduction. The Figure’s caption was edited to mention the characteristic C6-C2-C6 structure of stilbenoids. All other Figures were re-labeled accordingly.

Comment: Figure A2: Among the four donors, the desult of the bacterial composiotn of D2 showed a quite different with those of the others. Please describe the consideration on this result.

Response: In response to this comment, a consideration of the inter-individual variability of human gut microbiota was included (lines 300-307). Although not the focus of this paper, we nevertheless consider it a very important consideration when interpreting the results of this and similar studies, even if sample sizes are small.

Reviewer 2 Report

Molecules-438456. The authors present an original research report on the effect of six selected stilbenoids over the relative abundance of fecal microbial communities (16S rRNA gene sequencing) after 24h batch fermentation, using fresh fecal specimens obtained from four healthy donors. The authors provided enough evidence as to a substrate-specific on microbial changes at different taxonomic levels.

Major observations

1.     The authors mention in their experimental design section and along the manuscript that they used a batch-fermentation human-colon model when in fact they used an in vitro fecal fermentation (FFM) system. This observation does not diminish the value of all findings but, in light of current expert recommendations.

2.     Media composition detailed in sections 3.3.2. and 3.3.3. are not commonly used to simulate “colonic conditions”; in fact, your discussion (lines 242-245) provides an argument as to the “artificial” nature of the culture media. Authors should provide information as to how such chemical compositions mimic human colonic conditions. At least provide one or two relevant reports on this matter.

3.     Comparing 6 compounds with similar chemical features with one or more bioactivities, provides a unique opportunity to discuss the effect of specific chemical moieties (functional groups) on such bioactivities. You should re-inforce your discussion in this field.

4.     Section 2.4 should be included in the preceding section (results) and not as a separate discussion section.

5.     Subject D2 is so atypical than the other three and Control 0h is very different to control 24h in subjects D1 and D2 (eutrophic individuals).

6.     Authors should be aware that statistical analysis helps to avoid type 1 or 2 errors the leads to misleading on data interpretation. For example, line 88: “Although not statistically significant, the trend is clearly visible…” why is that?

Minor observations

·         Title. Please change “gut” for “fecal”

·         Lines 27-28. “The most common effect was a further decrease of the relative abundance compared to the control, thus accentuating an already existing trend”, which common effect?

·         Line 30. “…higher incidence of effects”, what do you mean?

·         Line 39. “from” not “of”

·         Line 49. “many” not “several”, “such as” instead of ‘that range from”

·         Lines 46-50. Change paragraph as follows: “In plants, phenolic compounds are usually conjugated to sugars, organic acids and macromolecules (e.g. dietary fiber and proteins) and most of them are not properly released and absorbed at the small intestine, reaching the colon for further microbial fermentation; at colonic level, they are fermented by the resident gut microbiota (GM)…”

·         Line 57. “microbial strains”

·         Line 60. Phenolic compounds are currently considered probiotics as in case of fermentable dietary fiber (Kawabata et al. Molecules 2019, doi:10.3390/molecules24020370; Gibson et al. 2017, https://www.nature.com/articles/nrgastro.2017.75)

·         Line 65. “Anti-bacterial adhesion effect”

·         Tables and Figures and related discussion. If statistical analysis do not sustain differences among compared groups, authors must eliminate any misleading argument along the manuscript

Author Response

Major observations

Comment: 1.     The authors mention in their experimental design section and along the manuscript that they used a batch-fermentation human-colon model when in fact they used an in vitro fecal fermentation (FFM) system. This observation does not diminish the value of all findings but, in light of current expert recommendations.

Response: As suggested, the name of the model used was corrected to in vitro fecal fermentation (FFM) system.

Comment: 2.     Media composition detailed in sections 3.3.2. and 3.3.3. are not commonly used to simulate “colonic conditions”; in fact, your discussion (lines 242-245) provides an argument as to the “artificial” nature of the culture media. Authors should provide information as to how such chemical compositions mimic human colonic conditions. At least provide one or two relevant reports on this matter.

Response: Our medium is a standard medium that has been previously used by Alan Crozier and the research group from Glasgow University. It has also been used in other fermentation studies where mimicking human colonic conditions was part of the methodology.  https://doi.org/10.1016/j.freeradbiomed.2009.07.031,

https://doi.org/10.1124/dmd.111.039651 and https://doi.org/10.1002/(SICI)1097-0010(199606)71:2<209::AID-JSFA571>3.0.CO;2-4. These have now been cited on the manuscript in section 3.3.2.

Comment: 3.     Comparing 6 compounds with similar chemical features with one or more bioactivities, provides a unique opportunity to discuss the effect of specific chemical moieties (functional groups) on such bioactivities. You should re-inforce your discussion in this field.

Response: A discussion regarding functional groups and their effect on bioactivities was added (lines 271-299). A related metabolomic study that our lab team is conducting using the same fecal samples from this study will shed more light on this concept, and that paper will refer to this one.

Comment: 4.     Section 2.4 should be included in the preceding section (results) and not as a separate discussion section.

Response: Section 2.4 has now been included in section 2.3. In order to improve the clarity of presentation of our results and observations, this section has been edited and re-organized into two sections: Decrease in Relative Abundance, Increase in Relative Abundance. Additions included a consideration of phenolic functional groups on their bioactivity, and of the concept of inter-individual variability. Excluded was a brief discussion of S. equolifaciens and A. equolifaciens since this specific species were not identified in our study.

Comment: 5.     Subject D2 is so atypical than the other three and Control 0h is very different to control 24h in subjects D1 and D2 (eutrophic individuals).

Response: : In response to this comment, a consideration of the inter-individual variability of human gut microbiota was included (lines 300-307). Although not the focus of this paper, we nevertheless consider it a very important consideration when interpreting the results of this and similar studies, even if sample sizes are small.

Comment: 6.     Authors should be aware that statistical analysis helps to avoid type 1 or 2 errors the leads to misleading on data interpretation. For example, line 88: “Although not statistically significant, the trend is clearly visible…” why is that?

Response: Lines 88-89 (now lines 96-97) have been reworded to be more precise and descriptive of an observed response rather than calling it a trend or a significant finding. Similarly, to avoid misleading language and data interpretation, changes were made throughout the manuscript.

Minor observations

Comment: Title. Please change “gut” for “fecal”

Response: Edited as suggested.

Comment: Lines 27-28. “The most common effect was a further decrease of the relative abundance compared to the control, thus accentuating an already existing trend”, which common effect?

Response: By common, we mean the most frequently observed effect in regards to relative abundance. Lines 27-28 have been reworded to be more precise and decrease ambiguity. “A frequently observed effect was a further decrease of the relative abundance when compared to the control.” To be consistent, similar language changes were made throughout the manuscript.

Comment: Line 30. “…higher incidence of effects”, what do you mean?

Response: This is meant to convey the following, and has been reworded as such: Observed effects were more frequently attributed to resveratrol and piceatannol, followed by thunalbene and batatasin III.” To be consistent, similar language changes were made throughout the manuscript.

Comment: Line 39. “from” not “of”

Response: Edited as suggested.

Comment: Line 49. “many” not “several”, “such as” instead of ‘that range from”

Response: Edited as suggested.

Comment: Lines 46-50. Change paragraph as follows: “In plants, phenolic compounds are usually conjugated to sugars, organic acids and macromolecules (e.g. dietary fiber and proteins) and most of them are not properly released and absorbed in the small intestine, reaching the colon for further microbial fermentation; at colonic level, they are fermented by the resident gut microbiota (GM)…”

Response: Edited as suggested.

Comment: Line 57. “microbial strains”

Response: Edited as suggested.

Comment: Line 60. Phenolic compounds are currently considered probiotics as in case of fermentable dietary fiber (Kawabata et al. Molecules 2019, doi:10.3390/molecules24020370; Gibson et al. 2017, https://www.nature.com/articles/nrgastro.2017.75)

Response: We reviewed these sources, and although they agree with pre-biotic like effects of phenolic compounds, and acknowledge the increasing evidence of it,  they are not officially considered prebiotics. As stated in Gibson et al. 2017, “Plant polyphenols constitute a class of compounds that can also meet the criteria of prebiotics, although far more studies in the target host are required.”

Comment: Line 65. “Anti-bacterial adhesion effect”

Response: Edited as “bacterial adhesion effect” (lines 62 and 66).

Comment: Tables and Figures and related discussion. If statistical analysis do not sustain differences among compared groups, authors must eliminate any misleading argument along the manuscript

Response: Language has been revised and some content excluded (for example, the exclusion of a discussion regarding S. equolifaciens and A. equolifaciens since this specific species were not identified in our study) in order to avoid misleading arguments and data interpretation. We tried to be as transparent and objective as possible since we believe that this study is of high scientific value despite the small sample size, and hope that it will be regarded as a pilot for a larger, and more substantial, study.  

Round 2

Reviewer 2 Report

Thank you for accepting all suggestions. Congratulations your manuscript is very interesting